L-cysteine transporter-PCR to detect hydrogen sulfide-producing Campylobacter fetus

http://orcid.org/0000-0003-3586-1524 Farace Pablo D. 1
Morsella Claudia G. 2
Cravero Silvio L. 1
Sioya Bernardo A. 1
http://orcid.org/0000-0002-1147-4485 Amadio Ariel F. 3
Paolicchi Fernando A. 2
Gioffré Andrea K. 1 gioffre.andrea@inta.gob.ar
1 Instituto de Agrobiotecnología y Biología Molecular (IABIMO), Instituto Nacional de Tecnología Agropecuaria (INTA), Consejo Nacional de Investigaciones Científicas y Tecnológicas (CONICET) , Buenos Aires , Argentina
2 Laboratorio de Bacteriología-Grupo de Sanidad Animal. Unidad Integrada INTA-Universidad Nacional de Mar del Plata , Balcarce, Buenos Aires , Argentina
3 Consejo Nacional de Investigaciones Científicas y Tecnológicas (CONICET), Estación Experimental Agropecuaria-INTA , Rafaela, Santa Fe , Argentina
Azevedo Vasco
Electronic publication date: 2019 Nov 5
Publication date: 2019
Volume: 7
Electronic Location ID: e7820
Received 2019 Apr 26; Accepted 2019 Sep 3
Copyright: © 2019 Farace et al.
Copyright year: 2019
Copyright holder: Farace et al.
License: This is an open access article distributed under the terms of the Creative Commons Attribution License, which permits unrestricted use, distribution, reproduction and adaptation in any medium and for any purpose provided that it is properly attributed. For attribution, the original author(s), title, publication source (PeerJ) and either DOI or URL of the article must be cited.
License URL: https://creativecommons.org/licenses/by/4.0/

Keywords: PCR, Sulfide production, Molecular differentiation, Bovine genital campylobacteriosis

Funding: National Agency of Promotion of Science and Technology ANPCyT PICT2015-1541 National Scientific and Technical Research Council CONICET PIP11220150100316CO National Agricultural Technology Institute INTA PNBIO-1131032 This work was supported by the National Agency of Promotion of Science and Technology (ANPCyT, project PICT2015-1541), the National Scientific and Technical Research Council (CONICET, project PIP11220150100316CO) and the National Agricultural Technology Institute (INTA, project PNBIO-1131032). The funders had no role in study design, data collection and analysis, decision to publish, or preparation of the manuscript.

==============================
Phenotypic differences between Campylobacter fetus fetus and C. fetus venerealis subspecies allow the differential diagnosis of bovine genital campylobacteriosis. The hydrogen sulfide production, for example, is a trait exclusive to C. fetus fetus and C. fetus venerealis biovar intermedius. This gas that can be biochemically tested can be produced from L-cysteine (L-Cys). Herein, we report a novel multiplex-PCR to differentiate C. fetus based on the evaluation of a deletion of an ATP-binding cassette-type L-Cys transporter that could be involved in hydrogen sulfide production, as previously described. A wet lab approach combined with an in silico whole genome data analysis showed complete agreement between this L-Cys transporter-PCR and the hydrogen sulfide production biochemical test. This multiplex-PCR may complement the tests currently employed for the differential diagnosis of C. fetus.

Introduction

Campylobacter fetus is best known as a major veterinary pathogen that has a detrimental effect on reproductive efficiency of herds. However, in humans, this bacterium can also cause intestinal illness and, occasionally, severe systemic infections and thus the products from cattle and sheep are suspected as sources of transmission (Wagenaar et al., 2014). The classification of C. fetus subspecies relies on clinical features, host specificity, and phenotypic traits. Despite technical limitations and variable success, hydrogen sulfide (H2S) production as well as tolerance to glycine and NaCl, selenite reduction and resistance to antibiotics are the available biochemical tests currently employed as differential diagnosis of C. fetus (OIE, 2018; Schulze et al., 2006).

Members of C. fetus have different tropism, as evidenced in veterinary practice and in the diagnosis. The subspecies C. fetus venerealis (Cfv) is restricted to the bovine reproductive tract, and is associated to the venereal disease bovine genital campylobacteriosis (BGC), whereas C. fetus fetus (Cff) is mainly intestinal and is usually related to sporadic abortion. To date, the bovine products are subjected to strict regulations by the World Organization for Animal Health (OIE) and must be tested for the presence of C. fetus subsp. venerealis before international trading (OIE, 2018). Therefore, its differentiation at the subspecies level is critical. The isolation of the bacteria can confirm BGC and subsequently biochemical tests can determine the particular different isolates. Among the biochemical tests, glycine resistance and hydrogen sulfide(H2S) production are two of the best biochemical performing tests. For example, Cff strains show 1% glycine resistance and produce H2S in L-cysteine (L-Cys) enriched media. By contrast, Cfv strains fail to grow in 1% glycine-containing media and to produce H2S (Véron & Chatelain, 1973). Hence, these traits allow their discrimination. A glycine-tolerant variant of Cfv (C. fetus venerealis biovar intermedius, Cfvi) are frequently isolated in some countries such as USA, UK, South Africa and Argentina, which complicates their accurate identification (Schmidt, Venter & Picard, 2010; Van Bergen et al., 2005; Iraola et al., 2013). A third-host associated subspecies, C. fetus subsp. testudinum, completes the list of subspecies of C. fetus. This subspecies has been isolated from reptiles and humans (Fitzgerald et al., 2014) and therefore would not be relevant for animal production.

In a previous wide genome association study, Van der Graaf-van Bloois et al. (2016a) described a recent diversification of mammalian C. fetus and implicated a genetic factor associated to H2S production. They described a deletion in an ATP-binding cassette-type L-Cys transporter in Cfv strains. The operon structure of this L-Cys transporter has five coding sequences and three of them code for different molecular components of the transporter: the ATP-binding protein, the permease, and the substrate-binding protein (locus tags CFF8240_RS03845, CFF8240_RS03850 and CFF8240_RS03855 in C. fetus 82-40 genome, respectively). This L-Cys importer could be part of the Class 3 ABC-transporters (Licht & Schneider, 2011) and in Cfv the permease and the extracellular binding domain coding genes are deleted. This deletion may impair the transporter assembly, affecting the up-take of L-Cys. This therefore could explain the impaired production of H2S from this amino acid in Cfv strains. On these bases, we aimed to develop a simple molecular technique for detecting the L-Cys transporter-deletion polymorphism with the main purpose of identifying H2S-producing C. fetus strains.

Materials and Methods

Campylobacter fetus isolates and bacterial culture

All the C. fetus isolates (n = 36) were obtained from bovine clinical samples at the Bacteriology Unit (EEA-INTA Balcarce, Argentina). Thirty of these clinical isolates were randomly-selected for this study. In addition, the strains Cfv 97/608, Cfv 98/25 and Cfvi 99/541 were also selected because of the availability of their whole genome sequences (Van der Graaf-van Bloois et al., 2014; Iraola et al., 2013) and three additional isolates were selected to perform whole genome sequencing (see below).

All the C. fetus isolates were grown on 7% blood-Skirrow selective agar plates (Oxoid, Hampshire, UK) with 1.25 IU/ml polymyxin B sulfate, 5 μg/ml trimethoprim, 10 μg/ml vancomycin and 50 μg/ml cycloheximide (Sigma-Aldrich, St. Louis, MO, USA). The plates were incubated under microaerophilic conditions (5% O2, 10% CO2 and 85% N2) for 72 h at 37 °C. C. hyointestinalis NCTC11562 and the field isolate C. sputorum 08/209 were grown under the same conditions. C. coli NCTC11353 and C. jejuni NCTC11392 were cultured on Blood-Columbia agar plates (Oxoid) under microaerophilic condition for 24 h at 42 °C.

Biochemical tests

The classification of the subspecies was performed following standard protocols (OIE, 2018): sodium selenite reduction, 3.5% sodium chloride resistance, 1% glycine tolerance and H2S production in 0.02% L-Cys enriched medium. We also tested 1.3%, 1.5% and 1.9% glycine tolerance. The isolates were identified as Cff if they reduced sodium selenite, produced H2S and showed sodium chloride tolerance and at least 1% glycine resistance.

DNA isolation

A rapid protocol (freeze-thaw cycles) was applied to obtain the DNA template as follows. A loopful of each culture was collected and resuspended in 250 µl of sterile deionized water. Two cycles of freeze and boiling (−80/95 °C) were performed and the cellular debris were discarded after a centrifugation step. Two µl of the supernatant was used as PCR-template. High quality genomic DNA was obtained using mini spin columns (NucleoSpin Tissue; Macherey-Nagel GmbH & Co., Duren, Germany). DNA quality was tested using the Qubit 4 fluorometer (Invitrogen, Carlsbad, CA, USA; Thermo Scientific, Waltham, MA, USA) and further used for sequencing purpose.

L-Cys transporter-PCR

One forward and two reverse primers (Fwd 5′-gtccatttacttatcacgataacagtgg-3′, Rev1 5′-gatattaggctaagaggaatggtgtattg-3′ and Rev2 5′-ctcccgtatctacatgaaagctaatatc-3′) were designed for a multiplex-PCR format using open source Unipro UGENE 1.31 (Okonechnikov et al., 2012) (Fig. 1A). The amplification mix consisted of 1 × GoTaq green Reaction buffer (1.5 mM MgCl2), 0.25 mM of each dNTP, 0.1 μM of each primer, and 1.25 U Taq polymerase (Promega Corp., Madison, WI, USA), nuclease-free water to reach a final volume of 25 µl and Campylobacter DNA template.

Figure 1 Differential L-Cys Transporter-PCR.

(A) Schematic representation of the organization of the genes encoding the L-Cys transporter in C. fetus. Primer targeting regions and expected PCR-products are shown. The gray arrow represents the permease protein YckJ coding gene (locus tag CFF8240_RS03850 in Cff 82-40 genome) which is deleted in Cfv strains. The light gray arrow represents the extracellular-binding protein YckK coding gene (locus tag CFF8240_RS03855), which is partially deleted in Cfv. ATP-binding protein coding gene (locus tag CFF8240_RS03845) which is another component of the transporter is conserved in both subspecies and is represented by black arrow. (B) Representative agarose gel electrophoresis. Lane 1, negative control (water); lane 2, Cff 08/421; lane 3, Cff 96/136; lane 4, Cfvi 06/341; lane 5, Cfv 97/608; lane 6, Cfvi 03/596 and lane 7, Cfv 95/258. Under the set conditions, the product of 1,390 bp is absent. M: molecular weight marker, 1 kb DNA ladder (Promega).

The touch-down amplification program consisted of an initial step at 94 °C for 3 min, 10 cycles at 94 °C for 1 min, followed by annealing temperatures starting at 55 °C for 1 min and decreasing 1 °C per cycle from 55 to 45 °C. Then, an extension step was performed at 72 °C for 1 min, followed by 30 cycles with an annealing at 51 °C, and a final termination step at 72 °C for 8 min.

Under these conditions, the absence of the expected product of 1,390 bp makes the interpretation of the PCR results easy. A product of 714-bp is indicative of Cff and Cfvi strains (which have a complete version of the operon and are H2S-producing strains), whereas a 310 bp product refers to Cfv strains (which contain a partly deleted operon and are non- H2S-producing strains). All the products were resolved in 1.5% agarose gel electrophoresis and visualized by ethidium bromide staining. The PCR-products were submitted to the UGB unit-INTA to confirm their identity through Sanger sequencing.

In silico-PCR: whole genome sequencing and genomic data analysis

We selected three isolates from bovine abortions (Cff 13/344, Cff 08/421 and Cfvi 06/341) of the most productive agricultural areas of Argentina. Paired-end Nextera XT libraries were constructed and sequenced in a MiSeq sequencer (2 × 250 pb, Illumina). A quality trimming step was applied to raw reads using Trimmomatic (Bolger, Lohse & Usadel, 2014). De novo assembly was done using SPAdes v3.11.1 (Bankevich et al., 2012). Contigs were oriented using Mauve (Darling et al., 2004; Rissman et al., 2009) and the genome of C. fetus venerealis 97-608 as a reference (NZ_CP008810.1). The genomes were annotated using PROKKA (Seemann, 2014) and RASTtk (Brettin et al., 2015). The assembly summary statistics is shown in Table S1.

In total, whole-genome sequence data of 214 C. fetus strains (Cff, n = 152; Cfv, n = 42; Cfvi, n = 19 and one strain not identified at the subspecies level, Cf = 1) from 19 countries and different hosts (bovine, n = 117; human, n = 78; ovine, n = 15; monkey, n = 1 and unknown, n = 3) were screened to search for the target sequences of the primers designed for the L-Cys transporter-PCR protocol. These data included the three genomes obtained in this study (Cff 13/344, Cff 08/421 and Cfvi 06/341) and 37 publicly available genomes from GenBank. Additionally, reads from C. fetus strains (n = 174) deposited in ENA database (https://www.ebi.ac.uk/ena/) were also assembled, as mentioned above, and subsequently analyzed as follows. The Primer map software (http://www.bioinformatics.org/sms2/primer_map.html) was used for global searching of Fwd, Rev1 and Rev2 primer sequences. Primer Map output is a textual map showing the annealing positions of PCR primers. Afterwards, several conditions were evaluated, including annealing of both primers of each pair and their orientation. The position of each target annealing site was employed to estimate the amplicon size. The program, by default, does not allow mismatches. Cases where the annealing was confirmed for a single primer were classified as not detected or unknown.

Statistics

The agreement between the H2S production biochemical test and the L-Cys transporter-PCR was tested with Cohen´s Kappa statistic.

Results

L-Cys transporter-PCR: wet-lab assay

The multiplexed PCR-based approach herein designed produced a differential band pattern between the C. fetus isolates with distinct H2S-biochemical test results (Fig. 1B). This protocol was named L-Cys transporter-PCR. We tested 36 biochemically typed isolates with this L-Cys transporter-PCR, followed by electrophoresis of the products in agarose gel to reveal the size of the amplicons. A single amplification product was obtained in all the tested strains. The retrieved band from Cff and Cfv biovar intermedius (Cfvi) strains was of 714 bp. This result coincided with a complete version of the L-Cys transporter operon and this pattern was named “CFF/CFVI.” Amplifications from Cfv strains generated a smaller product of 310 bp, equivalent to a partially deleted operon, and this profile was named “CFV” (Fig. 1B). This L-Cys transporter-PCR allowed a differential testing that avoided a negative result in presence of C. fetus DNA. Indeed, a negative result, sometimes could be indicative of both the absence of the specific target and the presence of inhibitors in the sample. As expected, no product was obtained from DNA of Campylobacter spp. other than C. fetus (C. hyointestinalis, C. coli, C. jejuni and C. sputorum) (Table 1). This result confirmed the specificity of this L-Cys transporter-PCR test for C. fetus.

Table 1 L-Cys transporter-PCR: analysis of Argentinian C. fetus isolates and Campylobacter spp. strains.

Strain	Origin	Biochemical test	Phenotypic
ID	L-cys transporter-PCR pattern	
1% Glycine resistance	H2S production	
Cff 96-136	Bahía Blanca, BA	+	+	Cff	CFF/CFVI	
Cff 08-421	Gral. López, SF	+	+	Cff	CFF/CFVI	
Cff 14-284	Pila, BA	+	+	Cff	CFF/CFVI	
Cff 04-240	Olavarría, BA	+	+	Cff	CFF/CFVI	
Cff 13-344	Balcarce, BA	+	+	Cff	CFF/CFVI	
Cff 11-572	Balcarce, BA	+	+	Cff	CFF/CFVI	
Cff 89-222	Balcarce, BA	+	+	Cff	CFF/CFVI	
Cff 90-189	Balcarce, BA	+	+	Cff	CFF/CFVI	
Cff CI N3	Balcarce, BA	+	+	Cff	CFF/CFVI	
Cff 01-165	Santa Rosa, LP	+	+	Cff	CFF/CFVI	
Cff 01-64	Balcarce, BA	+	+	Cff	CFF/CFVI	
Cff 05-622	Cnel. Dorrego, BA	+	+	Cff	CFF/CFVI	
Cff 11-262	Balcarce, BA	+	+	Cff	CFF/CFVI	
Cff 11-295	Saladillo, BA	+	+	Cff	CFF/CFVI	
Cff 11-360	Necochea, BA	+	+	Cff	CFF/CFVI	
Cff 11-685	Balcarce, BA	+	+	Cff	CFF/CFVI	
Cff 11-408	Necochea, BA	+	+	Cff	CFF/CFVI	
Cff btu5	BA	+	+	Cff	CFF/CFVI	
Cff btu6	BA	+	+	Cff	CFF/CFVI	
Cff btu7	BA	+	+	Cff	CFF/CFVI	
Cff 18-09	BA	+	+	Cff	CFF/CFVI	
Cff 18-100	BA	+	+	Cff	CFF/CFVI	
Cfv 97-608	Hucal, LP	−	−	Cfv	CFV	
Cfv 95-258	Mar Chiquita, BA	−	−	Cfv	CFV	
Cfv 08-382	Gral. Belgrano, BA	−	−	Cfv	CFV	
Cfv 05-355	Balcarce, BA	−	−	Cfv	CFV	
Cfv 98-25	Gral. Pueyrredón, BA	−	−	Cfv	CFV	
Cfv 19-01	BA	−	−	Cfv	CFV	
Cfvi 06-341	Pehuajó BA	−	+	Cfvi	CFF/CFVI	
Cfvi 03-596	Pehuajó, BA	−	+	Cfvi	CFF/CFVI	
Cfvi 02-146	BA	−	+	Cfvi	CFF/CFVI	
Cfvi 98-472	Azul, BA	−	+	Cfvi	CFF/CFVI	
Cfvi 99-541	Balcarce, BA	−	+	Cfvi	CFF/CFVI	
Cfvi 07-379	Mar Chiquita, BA	−	+	Cfvi	CFF/CFVI	
Cfvi 00-305	BA	−	+	Cfvi	CFF/CFVI	
Cfvi 03-596	Pehuajó, BA	−	+	Cfvi	CFF/CFVI	
C. sputorum 08-209	Balcarce, BA	ND	ND	ND	–	
C. coli NCTC11353	National Collection of Type Cultures, England	ND	ND	ND	–	
C. hyointestinalis NCTC11562	National Collection of Type Cultures, England	ND	ND	ND	–	
C. jejuni NCTC11392	National Collection of Type Cultures, England	ND	ND	ND	–	
Notes:

“CFF/CFVI pattern” means that all the components of the L-Cys transporter are present and therefore, a product of 714 bp is obtained. “CFV pattern” means that the L-Cys transporter is deleted and a product of 310 bp is obtained. “−” means that the amplification product was absent.

BA, Buenos Aires province; LP, La Pampa province; SF, Santa Fe province; ND, Not determined.

The results from the L-Cys transporter-PCR analysis displayed a perfect correlation with the H2S production test (κ = 1). The analysis of concordance between tests is shown in Table S2.

We also addressed an in silico analysis of genomic sequences from mammalian C. fetus to further support this conclusion.

L-Cys transporter-PCR : in silico screening

To study the performance of the L-Cys transporter-PCR in a more diverse panel of strains, we applied an in-silico PCR-strategy by performing searches of the primer targeting sequences in whole genomes of 214 C. fetus strains (three of which were obtained in this study by Next-Generation Sequencing technology). For this purpose, we employed the online Primer map application. The same products as the obtained by the wet lab-PCR were considered among all the predicted PCR products and the same patterns were determined according to the product size. This approach confirmed the primer annealing sites, and consequently, also allowed us to define the type of L-Cys transporter operon in 213 out of 214 C. fetus strains (Table 2). The target annealing sites were highly conserved because of the lack of nucleotide mismatches in these strains. The in silico-PCR was able to predict the annealing sites for Fwd-Rev2 primers in the genome of Cfv Azul-94 but the target sites were located into different contigs. The product size was difficult to estimate and consequently this strain had inconclusive results (Table 2).

Table 2 In silico-PCR: analysis of whole-genome sequence data.

ID	Strain	Host	Source	Country	Accession number		H2S production (Reference)	PCR L-Cys
transporter pattern	
Cff	04/554	Bovine	Foetus	AR	CP008808–CP008809	+	(Van der Graaf-van Bloois et al., 2014)	CFF/CFVI	
Cff	08/421	Bovine	Foetus	AR	SOOT00000000	+	(This study)	CFF/CFVI	
Cff	110800-21-2	Bovine	Bull	NL	LSZN00000000	+	(Van der Graaf-van Bloois et al., 2014)	CFF/CFVI	
Cff	13/344	Bovine	Foetus	AR	SOYX00000000	+	(This study)	CFF/CFVI	
Cff	82/40	Human	Blood	US	CP000487	+	(Van Bergen et al., 2005)	CFF/CFVI	
Cff	Cff 98/v445	Bovine	Bull	UK	LMBH00000000	+	(Van Bergen et al., 2005)	CFF/CFVI	
Cff	ATCC 27374	Ovine	Foetus (brain)	Unk.	MKEI00000000	+	(Van Bergen et al., 2005)	CFF/CFVI	
Cff	BT 10/98	Ovine	Unknown	UK	LRAL00000000	+	(Van Bergen et al., 2005)	CFF/CFVI	
Cff	NCTC10842	Unknown	Unknown	Unk.	LS483431	+	(Van Bergen et al., 2005)	CFF/CFVI	
Cff	B0042	Bovine	Feces	UK	ERR419595	+	(Van der Graaf-van Bloois et al., 2016a)	CFF/CFVI	
Cff	B0047	Bovine	Feces	UK	ERR419600	+	(Van der Graaf-van Bloois et al., 2016a)	CFF/CFVI	
Cff	B0066	Bovine	Feces	UK	ERR419653	+	(Van der Graaf-van Bloois et al., 2016a)	CFF/CFVI	
Cff	B0097	Bovine	Feces	UK	ERR419653	+	(Van der Graaf-van Bloois et al., 2016a)	CFF/CFVI	
Cff	B0129	Bovine	Feces	UK	ERR419637	+	(Van der Graaf-van Bloois et al., 2016a)	CFF/CFVI	
Cff	B0130	Bovine	Feces	UK	ERR419638	+	(Van der Graaf-van Bloois et al., 2016a)	CFF/CFVI	
Cff	B0131	Bovine	Feces	UK	ERR419639	+	(Van der Graaf-van Bloois et al., 2016a)	CFF/CFVI	
Cff	B0151	Bovine	Feces	UK	ERR419648	+	(Van der Graaf-van Bloois et al., 2016a)	CFF/CFVI	
Cff	B0152	Bovine	Feces	UK	ERR419649	+	(Van der Graaf-van Bloois et al., 2016a)	CFF/CFVI	
Cff	B0167	Bovine	Feces	UK	ERR460866	+	(Van der Graaf-van Bloois et al., 2016a)	CFF/CFVI	
Cff	B0168	Bovine	Feces	UK	ERR460867	+	(Van der Graaf-van Bloois et al., 2016a)	CFF/CFVI	
Cff	S0693A	Bovine	Feces	UK	ERR419284	+	(Van der Graaf-van Bloois et al., 2016a)	CFF/CFVI	
Cff	S0478D	Bovine	Feces	UK	ERR419653	+	(Van der Graaf-van Bloois et al., 2016a)	CFF/CFVI	
Cfvi	01/165	Bovine	Mucus	AR	CP014568–CP014570	+	(Van Bergen et al., 2005)	CFF/CFVI	
Cfv	84/112	Bovine	Genital secretion	US	HG004426–HG004427	−	(Van Bergen et al., 2005)	CFV	
Cfv	97/608	Bovine	Placenta	AR	CP008810–CP008812	−	(Van Bergen et al., 2005)	CFV	
Cfv	ADRI 1362	Bovine	Unknown	AR	LREX00000000	+	(Van der Graaf-van Bloois et al., 2014)	CFF/CFVI	
Cfv	ADRI513	Unknown	Unknown	AU	LRFA00000000	+	(Van der Graaf-van Bloois et al., 2014)	CFF/CFVI	
Cfv	B10	Bovine	Unknown	US	LRET00000000	−	(Van der Graaf-van Bloois et al., 2014)	CFV	
Cfv	CCUG 33872	Bovine	Abortion	CZ	LREU00000000	−/+	(Willoughby et al., 2005; Van der Graaf-van Bloois et al., 2014)	CFF/CFVI	
Cfv	CCUG 33900	Bovine	Abortion	FR	LREV00000000	−	(Van der Graaf-van Bloois et al., 2014)	CFV	
Cfv	LMG 6570	Bovine	Unknown	BE	LREW00000000	−	(Van Bergen et al., 2005)	CFV	
Cfv	NCTC 10354	Bovine	Mucus	UK	CM001228	−	(Van Bergen et al., 2005)	CFV	
Cfv	WBT 011/09	Unknown	Unknown	UK	LMBI00000000	+	(Van der Graaf-van Bloois et al., 2014)	CFF/CFVI	
Cfv	zaf3	Bovine	Foetus	SA	LREZ00000000	+	(Van der Graaf-van Bloois et al., 2014)	CFF/CFVI	
Cfv	zaf65	Bovine	Unknown	SA	LREY00000000	+	(Van der Graaf-van Bloois et al., 2014)	CFF/CFVI	
Cfvi	02/298	Bovine	Foetus	AR	LRVK00000000	+	(Van Bergen et al., 2005)	CFF/CFVI	
Cfvi	03/293	Bovine	Foetus	AR	CP0006999–CP007002	+	(Van Bergen et al., 2005)	CFF/CFVI	
Cfvi	03/596	Bovine	Foetus	AR	LRAM00000000	+	(Van Bergen et al., 2005)	CFF/CFVI	
Cfvi	06/341	Bovine	Foetus	AR	SOYW00000000	+	(This study)	CFF/CFVI	
Cfvi	92/203	Bovine	Placenta	AR	LRVL00000000	+	(Van Bergen et al., 2005)	CFF/CFVI	
Cfvi	97/532	Bovine	Mucus	AR	LRER00000000	+	(Van Bergen et al., 2005)	CFF/CFVI	
Cfvi/Cfv	98/25	Bovine	Foetus	AR	LRES00000000	+/−/−	(Van Bergen et al., 2005; Van der Graaf-van Bloois et al., 2016a; This study)	CFV	
Cfvi	99/541	Bovine	Prepuce	AR	ASTK00000000	+	(Van Bergen et al., 2005)	CFF/CFVI	
Cff	H1-UY	Human	Blood	UY	JYCP00000000	n.a		CFF/CFVI	
Cff	HC1	Human	Blood	UY	QJTR00000000	n.a		CFF/CFVI	
Cff	HC2	Human	Cerebrospinal fluid	UY	QJTS00000000	n.a		CFF/CFVI	
Cff	CIT01	Human	Peripheral blood culture	IR	RBHV00000000	n.a		CFF/CFVI	
Cfv	642-21	Bovine	Uterus	AU	AJSG00000000	n.a		CFF/CFVI	
Cfv	66Y	Bovine	Prepuce	CA	JPQC00000000	n.a		CFV	
Cfv	Azul-94	Bovine	Abortion	AR	ACLG00000000	n.a		Unknown	
Cfv	B6	Bovine	Vagina	AU	AJMC00000000	n.a		CFV	
Cfv	TD	Bovine	Prepuce	CA	JPPC00000000	n.a		CFV	
Cf	MMM01	Human	Sepsis	IN	JRKX00000000	n.a		CFF/CFVI	
Cff	99/801	Bovine	Prepuce	AR	ERS739235	n.a		CFF/CFVI	
Cff	00/398	Bovine	Foetus	AR	ERS739236	n.a		CFF/CFVI	
Cff	00/564	Bovine	Prepuce	AR	ERS739237	n.a		CFF/CFVI	
Cff	01/320	Bovine	Foetus	AR	ERS739238	n.a		CFF/CFVI	
Cff	01/210	Bovine	Vaginal mucus	AR	ERS739239	n.a		CFF/CFVI	
Cff	04/875	Bovine	Foetus	AR	ERS739242	n.a		CFF/CFVI	
Cff	05/394	Bovine	Foetus	AR	ERS739243	n.a		CFF/CFVI	
Cff	05/434	Bovine	Vaginal mucus	AR	ERS739244	n.a		CFF/CFVI	
Cff	06/340	Bovine	Prepuce	AR	ERS739245	n.a		CFF/CFVI	
Cff	07/485	Bovine	Vaginal mucus	AR	ERS739248	n.a		CFF/CFVI	
Cff	08/362	Bovine	Foetus	AR	ERS739249	n.a		CFF/CFVI	
Cff	10/247	Bovine	Prepuce	AR	ERS739250	n.a		CFF/CFVI	
Cff	10/445	Bovine	Prepuce	AR	ERS739251	n.a		CFF/CFVI	
Cff	11/360	Bovine	Foetus	AR	ERS739252	n.a		CFF/CFVI	
Cff	11/427	Bovine	Vaginal mucus	AR	ERS739253	n.a		CFF/CFVI	
Cff	14/270	Bovine	Foetus	AR	ERS739254	n.a		CFF/CFVI	
Cff	15/301	Bovine	Vaginal mucus	AR	ERS739255	n.a		CFF/CFVI	
Cfvi	02/146	Bovine	Foetus	AR	ERS739240	n.a		CFF/CFVI	
Cfvi	06/195	Bovine	Foetus	AR	ERS739246	n.a		CFF/CFVI	
Cfvi	07/379	Bovine	Foetus	AR	ERS739247	n.a		CFF/CFVI	
Cff	2006/367h	Human	Cerebrospinal fluid	FR	ERS672242	n.a		CFF/CFVI	
Cff	2006/479h	Human	Feces	FR	ERS672243	n.a		CFF/CFVI	
Cff	2006/588h	Human	Cerebrospinal fluid	FR	ERS672244	n.a		CFF/CFVI	
Cff	2006/621h	Human	Blood	FR	ERS672245	n.a		CFF/CFVI	
Cff	2006/649h	Human	Feces	FR	ERS672246	n.a		CFF/CFVI	
Cff	2008/170h	Human	Feces	FR	ERS672247	n.a		CFF/CFVI	
Cff	2008/568h	Human	Joint fluid	FR	ERS672248	n.a		CFF/CFVI	
Cff	2008/604h	Human	Feces	FR	ERS672249	n.a		CFF/CFVI	
Cff	2008/691h	Human	Cerebrospinal fluid	FR	ERS672250	n.a		CFF/CFVI	
Cff	2008/755h	Human	Blood	FR	ERS672251	n.a		CFF/CFVI	
Cff	2008/898h	Human	Blood	FR	ERS672252	n.a		CFF/CFVI	
Cff	2010/41h	Human	Feces	FR	ERS672253	n.a		CFF/CFVI	
Cff	2010/524h	Human	Kidney	FR	ERS672254	n.a		CFF/CFVI	
Cff	2010/1094h	Human	Blood	FR	ERS672255	n.a		CFF/CFVI	
Cff	2010/1119h	Human	Feces	FR	ERS672256	n.a		CFF/CFVI	
Cff	2010/1180h	Human	Blood	FR	ERS672257	n.a		CFF/CFVI	
Cff	2012/60h	Human	Feces	FR	ERS672258	n.a		CFF/CFVI	
Cff	2012/185h	Human	Blood	FR	ERS672259	n.a		CFF/CFVI	
Cff	2012/286h	Human	Blood	FR	ERS672260	n.a		CFF/CFVI	
Cff	2012/331h	Human	Blood	FR	ERS672261	n.a		CFF/CFVI	
Cff	2012/879h	Human	Feces	FR	ERS672263	n.a		CFF/CFVI	
Cff	2012/1045h	Human	Joint fluid	FR	ERS672264	n.a		CFF/CFVI	
Cff	2014/52h	Human	Cerebrospinal fluid	FR	ERS672265	n.a		CFF/CFVI	
Cff	2014/602h	Human	Blood	FR	ERS672266	n.a		CFF/CFVI	
Cff	2014/790h	Human	Blood	FR	ERS672267	n.a		CFF/CFVI	
Cff	2014/947h	Human	Blood	FR	ERS672269	n.a		CFF/CFVI	
Cff	2014/1097h	Human	Feces	FR	ERS672270	n.a		CFF/CFVI	
Cff	2007/123h	Human	Cerebrospinal fluid	FR	ERS672271	n.a		CFF/CFVI	
Cff	2009/56h	Human	Cerebrospinal fluid	FR	ERS672272	n.a		CFF/CFVI	
Cff	CF156	Human	Blood	TR	ERS672273	n.a		CFF/CFVI	
Cfvi	21-C0091-10-14_2	Bovine	Prepuce	UK	ERS672276	n.a		CFF/CFVI	
Cff	GTC _08732	Human	Cerebrospinal fluid	JP	ERS672218	n.a		CFF/CFVI	
Cff	GTC _11236	Human	Feces	JP	ERS672220	n.a		CFF/CFVI	
Cff	96-48	Human	Feces	JP	ERS672224	n.a		CFF/CFVI	
Cff	01-187	Human	Blood	JP	ERS672226	n.a		CFF/CFVI	
Cff	2004/103h	Human	Cerebrospinal fluid	FR	ERS672233	n.a		CFF/CFVI	
Cff	2004/199h	Human	Cerebrospinal fluid	FR	ERS672234	n.a		CFF/CFVI	
Cff	2004/359h	Human	Blood	FR	ERS672235	n.a		CFF/CFVI	
Cff	2004/362h	Human	Placenta	FR	ERS672236	n.a		CFF/CFVI	
Cff	2004/526h	Human	Feces	FR	ERS672237	n.a		CFF/CFVI	
Cff	2004/598h	Human	Blood	FR	ERS672238	n.a		CFF/CFVI	
Cff	2004/605h	Human	Feces	FR	ERS672239	n.a		CFF/CFVI	
Cff	2004/637h	Human	Joint fluid	FR	ERS672240	n.a		CFF/CFVI	
Cff	2006/222h	Human	Blood	FR	ERS672241	n.a		CFF/CFVI	
Cff	ID111063	Human	Blood	CA	ERS739225	n.a		CFF/CFVI	
Cff	ID117228	Human	Blood	CA	ERS739226	n.a		CFF/CFVI	
Cff	ID129038	Human	Blood	CA	ERS739227	n.a		CFF/CFVI	
Cff	ID131159	Human	Feces	CA	ERS739228	n.a		CFF/CFVI	
Cff	ID134381	Human	Feces	CA	ERS739229	n.a		CFF/CFVI	
Cff	ID136207	Human	Blood	CA	ERS739230	n.a		CFF/CFVI	
Cff	ID136551	Human	Blood	CA	ERS739231	n.a		CFF/CFVI	
Cff	ID136656	Human	Blood	CA	ERS739232	n.a		CFF/CFVI	
Cff	ID136706	Human	Blood	CA	ERS739233	n.a		CFF/CFVI	
Cff	ID132939	Human	Blood	CA	ERS739234	n.a		CFF/CFVI	
Cff	2975	Human	Blood	TW	ERS739256	n.a		CFF/CFVI	
Cff	923	Human	Blood	TW	ERS739257	n.a		CFF/CFVI	
Cff	7035	Human	Blood	TW	ERS739258	n.a		CFF/CFVI	
Cff	My5726	Human	Blood	TW	ERS739259	n.a		CFF/CFVI	
Cff	1592	Human	Blood	TW	ERS739260	n.a		CFF/CFVI	
Cff	1830	Human	Blood	TW	ERS739261	n.a		CFF/CFVI	
Cff	8468	Human	Blood	TW	ERS739262	n.a		CFF/CFVI	
Cff	0003304-2	Human	Blood	TW	ERS739263	n.a		CFF/CFVI	
Cff	2115	Human	Blood	TW	ERS739264	n.a		CFF/CFVI	
Cff	2819	Human	Blood	TW	ERS739265	n.a		CFF/CFVI	
Cff	5871	Human	Blood	TW	ERS739266	n.a		CFF/CFVI	
Cff	1666	Human	Blood	TW	ERS739267	n.a		CFF/CFVI	
Cff	9502	Human	Blood	TW	ERS739270	n.a		CFF/CFVI	
Cfv	800	Human	Blood	TW	ERS739271	n.a		CFF/CFVI	
Cff	8031708	Human	Blood	TW	ERS739272	n.a		CFF/CFVI	
Cff	8025552	Human	Blood	TW	ERS739273	n.a		CFF/CFVI	
Cff	3069482	Human	Blood	TW	ERS739274	n.a		CFF/CFVI	
Cfv	C1	Bovine	Prepuce	SP	ERS739275	n.a		CFV	
Cfv	C2	Bovine	Prepuce	SP	ERS739276	n.a		CFV	
Cff	C3	Bovine	Prepuce	SP	ERS739277	n.a		CFF/CFVI	
Cff	C4	Bovine	Prepuce	SP	ERS739278	n.a		CFF/CFVI	
Cff	C5	Bovine	Prepuce	SP	ERS739279	n.a		CFF/CFVI	
Cfv	C6	Bovine	Prepuce	SP	ERS739280	n.a		CFF/CFVI	
Cff	C7	Bovine	Prepuce	SP	ERS739281	n.a		CFV	
Cff	C8	Bovine	Prepuce	SP	ERS739282	n.a		CFF/CFVI	
Cff	C11	Bovine	Prepuce	SP	ERS739285	n.a		CFF/CFVI	
Cfvi	C12	Bovine	Prepuce	SP	ERS739286	n.a		CFF/CFVI	
Cff	C13	Bovine	Prepuce	SP	ERS739287	n.a		CFF/CFVI	
Cff	C14	Bovine	Prepuce	SP	ERS739288	n.a		CFF/CFVI	
Cff	C15	Bovine	Prepuce	SP	ERS739289	n.a		CFF/CFVI	
Cff	C16	Bovine	Prepuce	SP	ERS739290	n.a		CFF/CFVI	
Cff	C17	Bovine	Prepuce	SP	ERS739291	n.a		CFF/CFVI	
Cfv	C19	Bovine	Prepuce	SP	ERS739293	n.a		CFV	
Cff	C20	Bovine	Prepuce	SP	ERS739294	n.a		CFF/CFVI	
Cff	C21	Bovine	Prepuce	SP	ERS739295	n.a		CFF/CFVI	
Cfv	C22	Bovine	Prepuce	SP	ERS739296	n.a		CFV	
Cfv	C23	Bovine	Prepuce	SP	ERS739297	n.a		CFV	
Cfv	C24	Bovine	Prepuce	SP	ERS739298	n.a		CFV	
Cfv	C25	Bovine	Prepuce	SP	ERS739299	n.a		CFV	
Cfvi	C26	Bovine	Prepuce	SP	ERS739300	n.a		CFF/CFVI	
Cfv	C27	Bovine	Prepuce	SP	ERS739301	n.a		CFV	
Cfvi	C28	Bovine	Prepuce	SP	ERS739302	n.a		CFF/CFVI	
Cff	C29	Bovine	Prepuce	SP	ERS739303	n.a		CFF/CFVI	
Cfv	C30	Bovine	Prepuce	SP	ERS739304	n.a		CFV	
Cfvi	C31	Bovine	Prepuce	SP	ERS739305	n.a		CFF/CFVI	
Cfvi	C32	Bovine	Prepuce	SP	ERS739306	n.a		CFF/CFVI	
Cfvi	C33	Bovine	Prepuce	SP	ERS739307	n.a		CFF/CFVI	
Cfv	C34	Bovine	Prepuce	SP	ERS739308	n.a		CFF/CFVI	
Cfv	BS 201/02	Bovine	Prepuce	GE	ERS686632	n.a		CFV	
Cfv	BS 76/04	Bovine	Foetus	GE	ERS686633	n.a		CFV	
Cfv	BS 38/06	Bovine	Prepuce	GE	ERS686634	n.a		CFV	
Cfv	07BS020	Bovine	Prepuce	GE	ERS686635	n.a		CFV	
Cfv	08CS0024	Bovine	Prepuce	GE	ERS686636	n.a		CFF/CFVI	
Cfv	09CS0030	Bovine	Prepuce	GE	ERS686637	n.a		CFV	
Cfv	11CS0190	Bovine	Prepuce	GE	ERS686638	n.a		CFV	
Cfv	11CS0191	Bovine	Prepuce	GE	ERS686639	n.a		CFV	
Cfv	13CS0183	Bovine	Prepuce	GE	ERS686640	n.a		CFV	
Cfv	14CS0001	Bovine	Prepuce	GE	ERS686641	n.a		CFV	
Cff	BS 456/99	Ovine	Foetus	GE	ERS686642	n.a		CFF/CFVI	
Cff	BS 458/99	Bovine	Foetus	GE	ERS686643	n.a		CFF/CFVI	
Cff	BS 03/04	Bovine	Foetus	GE	ERS686644	n.a		CFF/CFVI	
Cff	BS 91/05	Bovine	Prepuce	GE	ERS686645	n.a		CFF/CFVI	
Cff	08CS0027	Bovine	Prepuce	GE	ERS686646	n.a		CFF/CFVI	
Cff	11CS0098	Ovine	Placenta	GE	ERS686648	n.a		CFF/CFVI	
Cff	12CS0302	Bovine	Prepuce	GE	ERS686649	n.a		CFF/CFVI	
Cff	13CS0001	Bovine	Prepuce	GE	ERS686650	n.a		CFF/CFVI	
Cff	13CS0373	Monkey	Feces	GE	ERS686651	n.a		CFF/CFVI	
Cff	001A-0374	Human	Blood	CA	ERS686652	n.a		CFF/CFVI	
Cff	001A-0648	Human	Blood	CA	ERS686653	n.a		CFF/CFVI	
Cff	LR133	Ovine	Foetus	NZ	ERS846544	n.a		CFF/CFVI	
Cff	1	Bovine	Prepuce	UK	ERS846553	n.a		CFF/CFVI	
Cff	2	Bovine	Prepuce	UK	ERS846554	n.a		CFF/CFVI	
Cff	3	Ovine	Placenta	UK	ERS846555	n.a		CFF/CFVI	
Cff	4	Ovine	Placenta	UK	ERS846556	n.a		CFF/CFVI	
Cff	5	Ovine	Placenta	UK	ERS846557	n.a		CFF/CFVI	
Cff	6	Bovine	Prepuce	UK	ERS846558	n.a		CFF/CFVI	
Cff	7	Ovine	Foetus	UK	ERS846559	n.a		CFF/CFVI	
Cff	8	Ovine	Foetus	UK	ERS846560	n.a		CFF/CFVI	
Cff	9	Ovine	Placenta	UK	ERS846561	n.a		CFF/CFVI	
Cff	12	Ovine	Placenta	UK	ERS846562	n.a		CFF/CFVI	
Cff	13	Bovine	Prepuce	UK	ERS846563	n.a		CFF/CFVI	
Cff	14	Ovine	Placenta	UK	ERS846564	n.a		CFF/CFVI	
Cff	15	Ovine	Placenta	UK	ERS846565	n.a		CFF/CFVI	
Cff	17	Ovine	Foetus	UK	ERS846566	n.a		CFF/CFVI	
Cfv	JCM_2528	Bovine	Vaginal mucus	UK	ERS846567	n.a		CFF/CFVI	
Cfv	161/97	Bovine	Prepuce	BR	ERS846568	n.a		CFF/CFVI	
Cfv	515/98	Bovine	Prepuce	BR	ERS846569	n.a		CFF/CFVI	
Notes:

“CFF/CFVI pattern” means that a complete L-Cys transporter is present. Hybridization of the primer pair Fwd-Rev1-template should occur and a product of 714 bp is predicted. “CFV pattern” means that the L-Cys transporter is partially deleted. Hybridization of the primer pair Fwd-Rev2-template should occur and a product of 310 bp is predicted.

Country code: US, United States; AR, Argentina; UK, United Kingdom; CZ, Czech Republic; FR, France; AU, Australia; CA, Canada; SA, South Africa; NL, The Netherlands; UY, Uruguay; BE, Belgium; IR, Ireland; IN, India; TR, Turkey; JP, Japan; TW, Taiwan; SP, Spain; GE, Germany; BR, Brazil. N.A: Not available.

The hydrogen sulfide production data were available for 43/214 of the studied strains, three from this study and forty from publicly available results (Van Bergen et al., 2005; Van der Graaf-van Bloois et al., 2014, 2016a; Willoughby et al., 2005). However, two of the evaluated strains have shown discrepant results according to the literature and were excluded from this analysis. Interestingly, all of the H2S-producing strains displayed a CFF/CFVI pattern, whereas the non- H2S-producing strains, with unequivocally results according to the biochemical test, presented a CFV pattern (k = 1) (Table 2). The analysis of concordance between in silico-PCR and H2S production is shown in Table S3.

Despite this concordance with the H2S-production test, 14 out of 43 strains that were identified as Cfv in the database did not match with the criteria of the deleted L-Cys transporter for this subspecies. Instead, these strains displayed a CFF/CFVI pattern (Table 2). This situation is also reflected by the overall analysis where the in silico study was able to assign the expected result in 92% (197/213) of the strains (one strain with inconclusive subspecies identification was excluded from the analysis). This partial discrepancy could be attributed to the different methods employed to determine subspecies and this information is not available for most of the strains used in this analysis.

As a proof of concept, we assessed six local field isolates (Cff 08-421, Cff 13-344, Cfv 97-608, Cfv 98-25, Cfvi INTA 99/541 and Cfvi 06-341) through the wet-lab and in silico-PCR approaches. The strains Cfv 97-608, Cfv 98-25 and Cfvi INTA 99/541 were selected from the C. fetus collection because of their genomic sequence availability (Van der Graaf-van Bloois et al., 2016a; Iraola et al., 2013). The L-Cys-transporter-PCR results perfectly matched the in silico-PCR predictions (Tables 1 and 2).

Altogether, this study showed a strong concordance between the results of the L-Cys transporter-PCR and the H2S-production test for C. fetus analysis. Furthermore, it highlights the lack of consensus in the classification of these bacteria between the different laboratories around the world.

Discussion

To date, phenotypic tests are among the most valuable methods to identify and differentiate microorganisms. However, these tests are usually time-consuming, because they are growth-rate dependent, and the whole process depends on the objectivity and skills of the operator. Furthermore, a proper standardization, which is essential to obtain reliable and reproducible results, is often absent. Despite all this, the replacement of these phenotypic tests by molecular techniques is not always an alternative to date. The antimicrobial resistance constitutes a good example of complementary testing, and this particular phenotypic trait can be tested by bacteriological methods and at molecular level in several pathogens (Fluit, Visser & Schmitz, 2001).

Over the last years, researchers have proposed many genotypic tests to facilitate C. fetus differentiation. For example, different studies have employed molecular techniques such as PCR based on different target genes to differentiate Cfv from Cff (Hum et al., 1997; Van Bergen et al., 2005; Abril et al., 2007). However, to date there is no clear consensus on the best method to assess C. fetus subspecies. The main problems rise from the limited number of tested strains, the failure to identify Cfvi strains and the low concordance with other techniques such as AFLP and, mainly, biochemical tests (Willoughby et al., 2005; Schulze et al., 2006; Schmidt, Venter & Picard, 2010).

A genome-wide association study has proposed the association between candidate gene loci coding for the L-Cys transporter and the H2S production, which together to glycine resistance is one of the phenotypic traits available for assessing C. fetus subspecies to date. According to this, H2S-producing C. fetus strains, commonly classified as Cff and Cfvi, have a complete L-Cys transporter operon, whereas the non-producing H2S C. fetus strains, classically classified as Cfv, have a deleted L-Cys transporter. It is important to mention that C. fetus subsp. testudinum (Cft), the last subspecies proposed of C. fetus, has a complete version of the operon. This is the case of the strain Cft 03/427 (whose genome is the representative of the species) which has been concordantly described as an H2S-producing strain elsewhere (Van der Graaf-van Bloois et al., 2016a). To date, this subspecies has not been described in cattle, and for this reason it was excluded of this study.

In this work, we have designed a multiplex-PCR protocol to provide a molecular tool to contribute to C. fetus characterization and differentiation. This L-Cys transporter-PCR showed an excellent correlation with the H2S production test according to both wet lab and in silico approaches. As other molecular techniques, this PCR failed to differentiate Cff from Cfvi strains. This will limit its use in countries where this biovar is prevalent. However, until more discriminative techniques are developed, its usefulness could be further enhanced by combining this technique with other complementary test, such as the glycine resistance assay.

In addition to practical implications of this tool in the laboratory, this study also contributes to the existing debate around C. fetus subspecies classification.

In this study, we tested C. fetus strains isolated and typed at the Bacteriology Unit of INTA-Balcarce (Argentina), which has a long history in culturing this bacterium and in performing its biochemical based classification. In this way, the wet-lab approach showed a perfect correlation not only with the H2S production test, but also with the C. fetus subspecies. Indeed, a CFF-CFVI pattern, which is indicative of L-Cys complete transporter, was associated with H2S-producing strains typically classified as Cff or Cfvi. By contrast, a CFV pattern, which is indicative of a deleted transporter, was exclusively associated with H2S-non-producing strains, which are typically classified as Cfv.

On the other hand, when we performed the in silico study, we analyzed genomic data from strains classified elsewhere by both molecular based approaches and/or biochemical tests and, as mentioned above, both techniques frequently displayed discordant results. In this way, we have obtained a perfect correlation with the H2S production test, but not with the reported subspecies of the strains.

This discordancy is well reflected by the strain 98-25. Researchers from the Bacteriology Unit of INTA-Balcarce isolated this strain in 1998 from aborted foetus, and originally typed it as Cfv because of its glycine sensitivity and its inability to produce H2S. This strain was included in this study and the PCR-L-Cys result was concordant with the phenotype of this strain. Later studies have also tested this strain and successfully sequenced its genome. Indeed, Van Bergen et al. (2005) typed it as glycine sensitive and H2S positive (typical traits of Cfvi strains). Later, Van der Graaf-van Bloois et al. (2014) reported it as non-H2S-producing strain. However, in this latter work, it has been called Cfvi 98-25 regardless the biochemical traits reported. In our study, the in silico sequence data analysis revealed the polymorphism of the L-Cys transporter (CFV pattern) typical of the non- H2S- producing strains.

Therefore, the same strain could display different biochemical traits when assayed in different labs -or time- and this is the classical bottle-neck of phenotypic tests.

We initially had other discrepancies with some isolates. Remarkably, 14 out of 43 Cfv genomic sequences tested in silico showed a complete version of the L-Cys transporter (CFF/CFVI pattern). Hence, at first glance, the hypothesis that all Cfv isolates do have a deleted L-Cys transporter appeared as not valid, according to the in silico analysis. However, when we searched the biochemical tests reported for some of these strains, we concluded that the in silico results presented here were concordant with the H2S production test. This discrepancy with the subspecies assigned could be due to the classification method of the strains that is frequently based on molecular techniques regardless the biochemical test results and moreover; the chosen method is not always specified (Iraola et al., 2017). Altogether, the in silico analysis also supports the hypothesis that states the occurrence of a deletion in the transporter operon in non-H2S producing strains, which are classified as Cfv according to biochemical methods.

As was mentioned earlier, it is important to highlight that the strains from databases are not typed by the same methodology and this fact is not always taken into account. Consequently, this could be problematic as our study showed. The most widespread molecular-based method is the multiplex-PCR described by Hum et al. (1997). This PCR targets the parA gene to identify Cfv strains. This transfer-associated gene is harbored in a pathogenicity island which encodes a Type 4 Secretion System (T4SS). Although the presence of a T4SS has been previously associated to Cfv strains (Gorkiewicz et al., 2010), it has also been demonstrated later that some Cff strains can harbor the T4SS and their related genes (Van der Graaf-van Bloois et al., 2016b). Furthermore, distinct phylogenetic analyses of C. fetus suggest that the current classification in subspecies must be redefined. A phylogenomic study based on the core genome have shown that the strains are divided in two clusters. While all the Cfv and Cfvi strains were grouped in one genome cluster, the Cff strains were equally distributed in both clusters (Van der Graaf-van Bloois et al., 2014). Additionally, a phylogenetic reconstruction based on the divergence acquired by recombination have also shown that Cfv and Cff strains share the same clade, which differs clearly from the clade of Cft strains of reptile origin (Gilbert et al., 2018). This emphasizes a real need to go further toward current C. fetus classification and identification, which will have a significant impact on the diagnostic practice. As mentioned above, although this issue has been addressed in the literature and genomic studies have broadened and strengthened our knowledge of this bacterium (Van der Graaf-van Bloois et al., 2014, 2016a; Iraola et al., 2017), a concerted action toward C. fetus subspecies classification and differentiation has been neglected. There are no molecular markers associated to tropism or virulence of Cfv that could help with a differential diagnosis of the BGC (Gilbert et al., 2018).

Consequently, in light of these evidences, more research is essential to determine, as a first step, whether the differential diagnosis should be promoted and, if so, to improve or replace definitively the tests currently available. Another point to consider is that, veterinary diagnostic laboratories from developing countries are often refractory to replace those methods that have proven be useful, even if they are not the most suitable ones. Because of that, the adoption of genetic or even genomic-based methods has been delayed. One possible reason is the cost related to each method. The second main reason is the time lapse it takes for scientific knowledge to reach end users. Interdisciplinary research combining genomics, biochemistry, epidemiology and the provision of updated information and training to end users could shed light on this matter in the near future.

Conclusions

Biochemical tests including tolerance to glycine and H2S production are currently recommended by the OIE (2018) for C. fetus subspecies differentiation and are still employed in laboratories around the world. Thus, a molecular tool linked to a phenotypic trait is a valuable tool that could be more accurate and less time consuming than the available phenotypic tests. Mutagenesis and functional studies are essential to associate definitely this putative L-Cys transporter with the H2S production. Meanwhile, this study shows that this transporter constitutes a good marker that is useful for detecting H2S-producing C. fetus. Future actions will be addressed to test the L-Cys-PCR in clinical samples to propose it not only as a typing method, but as a detection technique and, as a second phase of validation, to transfer this technology to other labs to test the reproducibility of the results.

Finally, this work provides a molecular tool linked to H2S production in C. fetus and supports the findings of the pioneering study of Van der Graaf-van Bloois et al. (2016a).

Supplemental Information

Supplemental Information 1 Whole Genome Sequencing of three local C. fetus Strains: Overall Assembly Statistics.

Click here for additional data file.

Supplemental Information 2 Analysis of agreement between the H2S production test and L-Cys-PCR.

Contingency table and calculation of the Cohen’s Kappa coefficient. Data set from Table 1 comprising 36 strains was included. The analysis showed a perfect agreement (ĸ = 1) ( https://idostatistics.com/cohen-kappa-free-calculator/ ).

Click here for additional data file.

Supplemental Information 3 Analysis of agreement between H2S production test and in silico L-Cys-PCR.

Contingency table and calculation of the Cohen’s Kappa Coefficient. Data set from Table 2 comprising 41 strains with reported-H2S production test was included in the analysis. The analysis showed a perfect agreement (ĸ = 1) (https://idostatistics.com/cohen-kappa-free-calculator/).

Click here for additional data file.

Supplemental Information 4 Genome sequences obtained in this study through NGS.

Click here for additional data file.

The authors are grateful to Dr. Julia Sabio y García for editorial revision and Dr. Martín Zumárraga for critical reading of the manuscript.

Additional Information and Declarations

Competing Interests

Author Contributions

DNA Deposition

Data Availability

The authors declare that they have no competing interests.

Pablo D. Farace performed the experiments, prepared figures and/or tables, approved the final draft.

Claudia G. Morsella performed the experiments, contributed reagents/materials/analysis tools, prepared figures and/or tables, approved the final draft.

Silvio L. Cravero analyzed the data, authored or reviewed drafts of the paper, approved the final draft, bibliographic searching to trace biochemical data.

Bernardo A. Sioya performed the experiments, prepared figures and/or tables, approved the final draft.

Ariel F. Amadio analyzed the data, contributed reagents/materials/analysis tools, authored or reviewed drafts of the paper, approved the final draft.

Fernando A. Paolicchi analyzed the data, contributed reagents/materials/analysis tools, authored or reviewed drafts of the paper, approved the final draft.

Andrea K. Gioffré conceived and designed the experiments, analyzed the data, authored or reviewed drafts of the paper, approved the final draft.

The following information was supplied regarding the deposition of DNA sequences:

The genome sequences and sequence read data are available at GenBank and at the NCBI Sequence Read Archive: Strain: 06/341, Assembly: GCA_008526355.1, WGS: SOYW00000000, BioSample: SAMN10523499, Run: SRR9668101;

Strain: 08/421, Assembly: GCA_008526335.1, WGS: SOOT00000000, BioSample: SAMN10523721, Run: SRR9668103;

Strain: 13/344, Assembly: GCA_008527615.1, WGS: SOYX00000000, BioSample: SAMN10518270, Run: SRR9668102

The following information was supplied regarding data availability:

Farace, Pablo Daniel (2019): PrimerMap_Output. figshare. Dataset. DOI 10.6084/m9.figshare.9880847.v1.

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
