# Peer review of "L-cysteine transporter-PCR to detect hydrogen sulfide-producing Campylobacter fetus"

_PeerJ, doi:10.7717/peerj.7820_

## Round 0.1 · original submission · Major Revisions

Please follow the comments from the referees.

Reviewer 1 ·

Basic reporting

The authors developed a new molecular method using PCR to characterize Campylobacter fetus strains based on a previously reported deletion that could be associated to the production of H2S, hence affecting this phenotype that is widely used to type C. fetus subspecies using biochemical tests. The manuscript is well-structured, self-contained and literature is correctly cited. Despite I think this is a relevant work to its specific field and deserves publication, I feel that some improvements in the experimental approach and the discussion would provide more robustness and validity to these findings.

Experimental design

Major comments:

- From the 38 strains included in the study, only 4 out of them were Cfv (10%). I think authors should include more Cfv strains to avoid this bias.

- To provide a more extensive evaluation of this novel method, authors can benefit from the large collection of publicly available whole-genome sequences of C. fetus (>200). For example, screening the L-Cys transporter polymorphism among these genomes that represent an international collection of strains from different parts of the world and isolated from bovine, ovine and human hosts, would improve the robustness of the proposed PCR assay.

- The authors focused on Cff/Cfv that are mammal-associated C. fetus subspecies. There is a third suspecies, C. fetus subsp. testudinum, that is found in humans and reptiles. Despite this represents a distant phylogenetic lineage to Cff/Cfv, the manuscript should include a comment if this L-Cys transporter polymorphism is present or not in Cft. This can be easily achieved by screening several publicly available Cft genomes.

Validity of the findings

Major comments:

- I think this study represents a valid alternative or complement to C. fetus phenotyping based on biochemical tests. However, recent studies have questioned the clinical relevance and validity of biochemical phenotyping of C. fetus subspecies. In particular, van der Graaf-van Bloois et al. (2014, 10.1128/JCM.01837-14) first describe an inconsistency of biochemical and genomic characterization in C. fetus. In this work, they revealed two main phylogenetic lineages and that strains that were phenotypically classified as
Cfv and Cfvi belonged to a single lineage, but the phenotypically classified Cff strains were dispersed among both genome clusters. I would like to see in the revised version of this manuscript, a more in-depth discussion including these aspects, specifically discussing why today biochemical tests are useful in the analysis of C. fetus epidemiology.

Additional comments

Minor comments:

- Sequencing and assembly statistics of C. fetus genomes sequenced during this study would be more clear if presented in a table or supplementary table than in the main text.

Reviewer 2 ·

Basic reporting

Well written, clear figures, clear tables.
Missing reads in the SRA. should be made available

Experimental design

Sufficient detail. Perhaps the PCR should be tested in more labs to validate the outcome.

Validity of the findings

no comment

Additional comments

The paper by Farace et al describes the development and validation of a PCR to replace the biochemical H2S test that distinguishes C. fetus venerealis from C. fetus venerealis biovar intermedius and C. fetus fetus. The manuscript is well written, but the usefulness of a test that distinguishes two pathogenic variants of C. fetus from each other is somewhat limited, although one could arque that a negative test would immediately identify the isolate as the pathogenic subspecies venerealis instead of the presumed less pathogenic C. fetus fetus subspecies. Unfortunately the prevalence of the H2S positive C. fetus venerealis biovar intermedius is quite high in Argentina.

I have a couple of comments:

1. Please make the reads available as well on the SRA
2. Discuss the (limited) use of the test in countries with high prevalence of Cfvi
3. In how many labs was the PCR tested? Should it be the subject of a ring test? Discuss

---

## Round 0.2 · Minor Revisions

Please make the corrections of the reviewer and send us a new version.

Reviewer 2 ·

Basic reporting

Improved discussion. Well written

Experimental design

I like the addition of the in silico data. Although the test has some limitations (see below) in certain countries, it is still useful.

Validity of the findings

I would like to see the sentence "This might limit its use in countries where this biovar is present." Changed to "This will limit its use in countries where this biovar is prevalent", as that is the actual situation, it is not a hypothetical situation which may occur.

Additional comments

I have just one comment, see point 3, validity of the findings.

---

## Round 0.3 · accepted · Accept

Congratulations - your article has been accepted.